# Molecular Approaches to Treating Chronic Obstructive Pulmonary Disease: Current Perspectives and Future Directions

**DOI:** 10.3390/ijms26052184

**Published:** 2025-02-28

**Authors:** Sheryl-Phuc Vu, Kaleb Veit, Ruxana T. Sadikot

**Affiliations:** 1Division of Pulmonary, Critical Care & Sleep, Department of Internal Medicine, University of Nebraska Medical Center, Omaha, NE 68198, USA; shvu@unmc.edu (S.-P.V.); kveit@unmc.edu (K.V.); 2Department of Pathology and Microbiology, College of Medicine, University of Nebraska Medical Center, Omaha, NE 68198, USA; 3VA Nebraska Western Iowa Health Care System, Omaha, NE 68105, USA

**Keywords:** COPD, molecular mechanisms, therapy, inflammation, oxidative stress

## Abstract

Chronic obstructive pulmonary disease (COPD) is a clinical syndrome that presents as airflow limitation with poor reversibility accompanied by dynamic hyperinflation of the lung. It is a complex disease with chronic inflammatory airway changes caused by exposure to noxious particles or gases, such as cigarette smoke. The disease involves persistent inflammation and oxidative stress, perpetuated by frequent exacerbations. The prevalence of COPD is on the rise, with the prediction that it will be the leading cause of morbidity and mortality over the next decade. Despite the global burden of COPD and its associated morbidity and mortality, treatment remains limited. Although the understanding of the pathogenesis of COPD has increased over the last two decades, molecular approaches to develop new therapies for the treatment of COPD have lagged. Here, we review the molecular approaches that have the potential for developing novel therapies for COPD.

## 1. Introduction

Chronic obstructive pulmonary disease (COPD) is “a heterogeneous lung condition characterized by chronic respiratory symptoms (dyspnea, cough, expectoration) due to persistent abnormalities of the airways (bronchitis, bronchiolitis) and/or alveoli (emphysema) that often results in progressive airflow limitation” [1]. It affects more than 384 million people worldwide [2], and it is the third leading cause of death in the global population [3].

Cigarette smoking is the most well-known risk factor for the development of COPD [4]. Cigarette smokers lose lung function (decline in forced expiratory volume in 1 second (FEV1)) at a faster rate than nonsmokers, which improves with smoking cessation [5]. Furthermore, chronic bronchitis is more prevalent in smokers and improves with smoking cessation [6].

About 50% of global COPD cases are caused by non-tobacco-related risk factors [7]. Genetic risk factors, most notably mutations in the SERPINA 1 gene causing alpha1-antitrypsin deficiency [8], are associated with an increased susceptibility to COPD; other genetic risk loci have also been identified, such as CHRNA3, FAM13A, HHIP, and RIN3 [9]. Inhalational exposures to biomass fuels, such as wood and charcoal smoke, are associated with the development of COPD [10]. Notably, approximately 3 billion people worldwide use open fires for cooking and heating [11], representing a large at-risk population. Replacing biomass as a fuel source with cleaner fuels is associated with a reduction in the risk of COPD [12]. Air pollution increases the rate of lung function decline and the risk of COPD [13], and exposure to air pollutants increases the risk of COPD exacerbations [14]. Finally, the fraction of COPD attributable to occupational exposures is estimated to be 31.1% in nonsmokers [15]. Despite the global burden of COPD and its associated morbidity and mortality, treatment remains limited. This has led to an interest in molecular approaches to treating COPD, which is discussed below (Figure 1).

## 2. Current Pharmacologic Therapies

At present, pharmacologic treatment of COPD (Table 1) focuses on reducing symptoms and reducing the risk of exacerbations [16]. Non-pharmacologic approaches include smoking cessation, avoiding exposures to irritants, pulmonary rehabilitation, and management of co-morbidities. Pharmacologic approaches include long-acting beta-agonists (LABAs) and long-acting muscarinic antagonists (LAMAs). These improve dyspnea and health status and reduce exacerbation [17,18]. Studies of pharmacotherapy in COPD have also used changes in FEV1 values over time as a surrogate for the disease course [1]. Pharmacologic therapy with inhaled bronchodilators reduces the decline in FEV1 in patients with COPD [19]. Interestingly, only two studies of pharmacologic therapy in COPD have shown a mortality benefit: the use of triple therapy with a LABA, LAMA, and inhaled corticosteroid (ICS) reduces mortality in COPD patients based upon the results of the IMPACT and ETHOS studies [20,21]. Post hoc analysis suggested that the mortality benefit seen in these two studies may have been related to prior ICS discontinuation [22]. All other therapies with a mortality benefit in COPD are non-pharmacologic: smoking cessation [23], pulmonary rehabilitation [24], long-term oxygen therapy [25,26], non-invasive positive pressure ventilation [27], and lung volume reduction surgery [28].

Roflumilast is a phosphodiesterase 4 (PDE4) inhibitor used as an adjunct in the treatment of COPD patients with severely reduced lung function (FEV1 < 50%), a chronic bronchitis phenotype, and a history of exacerbations. Its mechanism of action is the reduction in airway inflammation through the cyclic adenosine monophosphate (cAMP) pathway [29]. Roflumilast use is associated with improvements in lung function [30] and a reduction in moderate to severe COPD exacerbations [31]. However, Roflumilast can be related to weight loss, depression, gastrointestinal symptoms, and insomnia [32].

Macrolide antibiotics have various anti-inflammatory and immunomodulatory effects on lung diseases, including COPD [33]. In patients with COPD with an increased risk of exacerbation, azithromycin has been shown to reduce the frequency of COPD exacerbations when taken daily (250 mg) or three times a week (500 mg) for one year [34,35]. When taken twice daily for one year, Erythromycin reduced the risk of COPD exacerbation in a study of COPD patients, 35% of whom had three or more exacerbations in the prior year [36]. Long-term azithromycin use can be associated with hearing loss, prolonged QT interval, and bacterial resistance [34]. With the limitations of existing therapies, new approaches are much needed to treat COPD.

## 3. Phosphodiesterase Inhibitors

Phosphodiesterase inhibitors are a class of medications that have been used in the management of COPD (Table 2) [16]. Theophyllines are nonspecific phosphodiesterase inhibitors that have been used as weak bronchodilators. By inhibiting this enzyme, cyclic adenosine monophosphate (cAMP) levels increase in inflammatory cells, reducing inflammation. Theophyllines also had other effects, such as weak respiratory center stimulation, decreased respiratory muscle fatigue, and mild diuretic properties. However, because of the side effects and drug interactions, these drugs had limited utility. Specific phosphodiesterase inhibitors are being developed, such as PDE-4 inhibitors that target the enzyme phosphodiesterase-4, which plays a role in the inflammatory process. Roflumilast is a current FDA-approved phosphodiesterase four inhibitor that reduces airway inflammation and relaxes the smooth muscle. These drugs have been shown to decrease the frequency and severity of exacerbations in patients with COPD, although their bronchodilator effect is limited [16,37]. Furthermore, these drugs have potential side effects, as mentioned above, which may limit their use. Inhaled PDE4 inhibitors such as Tanimilast may provide therapeutic benefits while limiting the gastrointestinal side effects [38].

Ensifentrine is a novel inhaled phosphodiesterase 3 (PDE3) and PDE4 inhibitor with both bronchodilator and anti-inflammatory properties. It was recently approved by the United States Food and Drug Administration for maintenance treatment of COPD. In the ENHANCE-1 and ENHANCE-2 studies, Ensifentrine significantly improved lung function and reduced the rate of moderate or severe COPD exacerbations, both when used as a monotherapy and when added to a LAMA, LAMA/ICS, LABA, or LABA/ICS. Notably, Ensifentrine has not been studied as an add-on to LAMA/LABA and LAMA/LABA/ICS combinations. The rate of adverse events with Ensifentrine was similar to placebo [39]. In general, the phosphodiesterase inhibitor’s role is preventive rather than symptomatic, focusing on reducing exacerbations and long-term inflammation.

## 4. Targeting Inflammation

Despite the addition of therapies such as phosphodiesterase inhibitors and macrolides to inhaled bronchodilators, many patients continue to remain symptomatic and experience exacerbations. COPD pathogenesis, onset, and development are orchestrated by the downstream consequences of infiltration from inflammatory cells, such as CD8+/CD4+ T lymphocytes, activated macrophages, eosinophils, and neutrophils. Chemical stress from harmful smoke burdens the airway and triggers the epithelium lining to produce cytokines (Interleukin 6, Interleukin 8, TNF-α, etc.) and chemokines [40,41]. Monocyte chemotactic protein (MCP-1, CCL-2) and macrophage inflammatory protein-1α (MIP-1α, CCL-3) are CC-chemokines, which act as chemo-attractants for inflammatory cells like macrophages and lymphocytes [42]. Neutrophil-derived CCL-2 and CCL-3 are involved in macrophage recruitment into inflamed tissue [42]. These inflammatory markers aggravate the epithelium lining and damage the airway tissue. Some anti-inflammatory options used are Azithromycin and phosphodiesterase 4 inhibitors [43]. Other anti-inflammatory biologics are currently utilized or studied for COPD treatment (Table 3).

### 4.1. Steroid

Another standard COPD treatment is corticosteroids and bronchodilators, which alleviate the inflammation and swelling of the airway. Despite being the common standard of care treatment, corticosteroids do not address all the components of the underlying inflammatory process. Patients with elevated neutrophilic inflammation are typically resistant to corticosteroid therapy [44]. In addition, several mechanisms of resistance to corticosteroids have been uncovered, including reduced activity of histone deacetylases related to an increase in oxidative stress [45,46].

Tumor Necrosis Factor-α (TNF-α) is correlated with smoking history and has been found to be elevated in COPD patients’ bronchioalveolar fluid samples. TNF-α modulates neutrophil chemotaxis and corticosteroid tolerance. An observational trial with TNF-α inhibitor showed a reduction in COPD exacerbation and a potential synergy with corticosteroid therapy [47].

### 4.2. Neutrophilic Inflammation

Neutrophilic inflammation is a significant contributor to the pathogenesis of COPD. Neutrophils are involved with phagocytosis, bacterial killing, degranulation, and the formation of neutrophil extracellular traps. Airway neutrophilia is exacerbated in bacterial and viral infections in patients with COPD. Neutrophil extracellular traps (NETs) are weblike scaffolds containing double-stranded DNA, neutrophil myeloperoxidases, and elastases [55]. Although NETs are formed as a host defense mechanism, releasing elastases and reactive oxygen species contributes to the perpetuation of inflammation. COPD exacerbations are associated with excessive NET production, worsening the airflow limitation. In a murine model of viral exacerbated COPD, Katsoulis et al. have shown that pharmacological inhibition of NETs attenuates inflammation while restoring lung function [48].

A clinical trial of an inhaled inhibitor PI3Kδ showed improved disease recovery in COPD patients [56]. Another strategy that targets neutrophil activity is the modulation of chemokines that recruit neutrophils to the lung microenvironment. Jasper et al. (2019) reported clinical trials on CXCR2 agonists and LTB4 receptor antagonists in airway diseases, with adverse events reported [57]. The current treatment for neutrophilic inflammation involves the use of macrolides such as Azithromycin [56]. Developing therapies that target NET inhibition needs to be studied in patients with COPD exacerbation.

### 4.3. Eosinophilic Inflammation

Besides neutrophilic inflammation, eosinophilic inflammation is relevant to COPD development [58]. As discussed below, biologics that target interleukins have anti-inflammatory effects. Mepolizumab has been shown to inhibit eosinophil-related exacerbation by orchestrating a pro-inflammatory cytokine, Interleukin 5. Itepekimab is currently under phase 3 clinical trial after a phase 2 clinical trial reported improved lung functions compared with the control group [53]. FDA-approved Benralizumab, Dupilumab, and Lebrikizumab also reduced COPD exacerbation by targeting the IL-5, IL-4, and IL-13 activities, respectively [43,49,58].

Precision therapy can target specific inflammatory pathways to target the cellular inflammation that triggers COPD exacerbation. This has resulted in the development of several biologic therapies, which are discussed below.

### 4.4. Biologics Targeting the Inflammation for the Treatment of COPD

Biologics are designed to target specific pathways involved in disease processes. Their use in COPD is being explored for patients with severe disease and features that overlap with asthma, particularly those with high levels of airway inflammation. Biologics are not yet standard therapy for COPD but are being studied for severe cases with specific inflammatory phenotypes. They may be considered in patients with frequent exacerbations despite optimal inhaler therapy and significant inflammatory markers [59].

The different endotypes of COPD have defined the predominance of neutrophilic vs. eosinophilic subtypes. Patients with eosinophilic signatures respond to steroids and have been studied to define the role of anti-eosinophilic biologics. Therefore, COPD patients with elevated blood eosinophil counts may respond to biologics targeting eosinophils or associated pathways such as Mepolizumab (anti-IL-5), which reduces eosinophil activity, and Benralizumab (anti-IL-5 receptor) that can deplete eosinophils and basophils. IL-4 and IL-13 pathways are targeted by Dupilumab (anti-IL-4 receptor). Dupilumab attenuates type 2 inflammation and is being investigated for COPD with mixed inflammatory phenotypes [50,59].

Biologic treatment of patients with eosinophilic COPD and frequent exacerbations has been studied using anti-interleukin-5 monoclonal antibodies (Mepolizumab) and anti-interleukin-5 receptor-alpha antibodies (Benralizumab), yielding mixed results [51,52]. Treatment of eosinophilic COPD in patients with a chronic bronchitis phenotype and frequent exacerbations with the humanized anti-interleukin-4 receptor alpha monoclonal antibody (Dupilumab) is associated with a reduction in exacerbations, improved FEV1, and improvement in symptoms and health-related quality of life [59]. Biologics targeted to eosinophils may reduce exacerbations, particularly in patients with high eosinophil counts. They may improve symptoms in subsets of patients with asthma-COPD overlap syndrome, reducing the need for oral corticosteroids.

The epithelial-derived alarmin cytokines IL-33 and TSLP have been implicated in certain patients with COPD [60,61]. Murine models of IL-33 gain of function have shown exaggerated airway inflammation with infiltration of eosinophils and increased mucin production with epithelial and goblet cell hypertrophy. Smoke exposure models in mice have confirmed an elevated level of IL-33 in the airways with attenuation of inflammation with inhibition of IL-33 using antibodies. Patients with COPD have been shown to have elevated levels of IL-33 [60,62], and genetic variations have shown a protective role for IL-33 loss of function in humans [53]. IL-33 antibodies (Tozorakimab and Tezepelumab) and receptor antagonists (Astegolimab) are undergoing clinical trials and will define their role in COPD management [54].

Thus, newer biologics hold promise for more targeted treatments; however, it is noteworthy that biologics may not benefit all COPD patients. Biomarkers like eosinophil levels or other biomarkers may help in patient selection. Furthermore, biologics are expensive, which may limit accessibility.

## 5. Small-Molecule Inhibitors

Drugs targeting specific kinases involved in inflammatory signaling (like Janus kinase (JAK) inhibitors) are being explored in COPD (Table 4). JAK inhibitors are a class of medications that target the JAK-STAT signaling pathway, which plays a crucial role in the immune response and inflammatory processes [63,64,65,66]. By inhibiting the JAK-STAT pathway, JAK inhibitors could potentially reduce inflammation and attenuate cytokine production levels, such as levels of cytokines like IL-6, TNF-α, and others involved in COPD. Thus, reduction in inflammation may mitigate exacerbations and reduce the frequency and severity of acute COPD exacerbations triggered by inflammation and would result in improved lung function. While these inhibitors are primarily approved for conditions like rheumatoid arthritis, psoriasis, and inflammatory bowel disease, there is growing interest in their potential application in COPD. Preclinical studies have demonstrated that JAK inhibitors can reduce lung inflammation and tissue damage in COPD models.

LAS194046, a novel pan-Janus kinase (JAK) inhibitor has been investigated in vitro in inflammatory cells from patients with COPD. Results of these studies showed that a combination of LAS194046 and fluticasone had anti-inflammatory and antioxidant effects [67]. There are some kinase inhibitors, such as Dilmapimod and Lormapimod, that have been used in COPD with no significant efficacy. Some JAK inhibitors, such as Tofacitinib and Ruxolitinib, are being investigated for their effects on COPD outcomes. However, these trials are in early phases, and the results are still pending or inconclusive.

JAK inhibitors can suppress the immune system, increasing the risk of infections, a significant concern for COPD patients already prone to respiratory infections. Identifying patients most likely to benefit from JAK inhibitors remains challenging and requires rigorous safety assessments. JAK inhibitors hold promise as a novel therapeutic approach for COPD, particularly for inflammation-driven phenotypes. However, more research and clinical evidence are needed to establish their safety and efficacy in this context.

## 6. Antioxidant Therapies

Cigarette smoking is one of the primary causes of the development of COPD. Upon initial exposure to cigarette smoke, the cells lining the airway and lungs undergo smoke-induced oxidative stress, which triggers prolonged inflammation even after smoking cessation [68]. Cigarette smoke carries a large volume of soluble oxidants that reduce oxygen to superoxide radicals. These free radicals and other reactive oxygen species (ROS) initiate a cascade of immune responses involving macrophages and neutrophils. These immune cells, in turn, release superoxide and hydrogen peroxide, which further exacerbate oxidative stress and inflammation in the lung tissues. Downstreamconsequences of oxidative stress are not limited to gene alteration, cell degranulation, phagocytosis decrease, infection susceptibility, and tissue damage [68]. The body has several antioxidant mechanisms to combat oxidative stress. Particularly in the lung, the imbalance between oxidant-antioxidant and dysregulation of NK-kB signaling contributes to COPD development and progression [69].

Nicotinamide Adenine Dinucleotide Phosphate (NADPH) oxidase (NOX) is a family of enzymes that generate ROS by transferring electrons to oxygen through the plasma membrane [70]. The superoxide anions produced by NOX can be converted rapidly into hydroxyl radicals and/or H_2_O_2_ [71]. It is reported that NOX1, NOX2, NOX4, and NOX5 are active in patients with end-stage COPD, while NOX1 and NOX4 are primarily detected in smoke-induced COPD [72]. Given the central role of oxidative stress in COPD, antioxidant therapies at the molecular scope are designed to restore the oxidation balance in the lung and airway tissues (Table 5).

It has been found that thiol-based antioxidants are an effective complement to the COPD standard-of-care regimen [68,69]. N-acetyl-L-cysteine (NAC) is efficacious in improving mucociliary clearance and reducing inflammation. Prolonged exposure to smoke significantly reduces glutathione levels, a major thiol antioxidant [68,74]. NAC increases cysteine’s stability and absorption for the synthesis of glutathione. Notably, NAC is effective at very low doses and suitable for prolonged treatment of chronic disease. Its effectiveness as a mucoactive drug and a potent antioxidant makes NAC a safe and cost-effective drug for COPD [73].

In addition to NAC, Procysteine is another thiol-based antioxidant and a cysteine donor that improves the synthesis of cellular glutathione [74]. The drug Erdostein has mucoactive and antioxidant properties and is prescribed twice a day for patients with prolonged COPD [75]. Erdostein and Procysteine, when used together, have been shown to reduce ROS production and inflammation cytokines in smoker patients with COPD [75].

Carbocysteine is a mucolytic antioxidant compound that increases sialomucin content while significantly reducing the levels of pro-inflammatory Interleukin 6 [75]. Extended treatment with Carbocystein minimizes the number of exacerbations in COPD patients and reduces their hospitalization time [75]. Similarly, Fudosteine, a modified version of NAC, has greater bioavailability that can lead to an increase in cellular cysteine, reduced mucin production, and reduced goblet cell hyperplasia. The favorable result is from animal research, and there are no current clinical trials [68,75].

Another antioxidant in COPD management involves Agonists of Nuclear factor erythroid 2-related factor 2 (Nrf2). Nrf2 is a transcription factor that regulates the expression of antioxidant genes, including those involved in glutathione synthesis. Nrf2 significantly increases glutathione synthesis and concert antioxidant response [68]. Nrf2 loss-of-function led to the development of emphysema-dominant COPD [77]. Several Nrf2 agonists, such as CDDO-Imidazolide, sulforaphane, and chalcones, have been developed with favorable results in pre-clinical studies [68]. Clinical studies are still needed to assess their safety and effectiveness in human populations.

Another key antioxidant defense strategy is facilitated by Glutathione Peroxidases (GPXs), which is a group of 8 enzymes that catalyze hydrogen peroxide and lipid hydroperoxide reduction [68,78]. It is found that GPXs activity was significantly lower in patients with severe COPD compared to patients with moderate COPD [79]. A recent study has determined the specific homolog GPx-1 is significantly reduced in COPD patients compared to non-COPD patients [78]. GPXs in red blood cells have also been found to be relevant to COPD patients with unfavorable pulmonary functions [68]. Ebselen, a mimic of glutathione Peroxidase-1, protects against inflammation resulting from cigarette smoke. Mainly, preclinical studies found Ebselen to avert smoke-induced gastrointestinal dysfunction [68,76].

Another compound of interest is Thioredoxin (TRx), a redox sensor that reduces oxidized proteins and enhances autophagy. TRx was found to improve NF-kB activity by preventing it from translocating to the nuclease and subsequently activating inflammatory genes [69]. TRx modulates oxidative stress in bronchial epithelial cells, type II pneumocytes, and macrophages and participates in the oxidation surveillance of several respiratory illnesses, including COPD [69]. Despite its therapeutic potential, it has yet to be evaluated in clinical trials for COPD.

In addition to pharmacological antioxidant therapies, other agents such as Celastrol and hydrogen are being investigated for their potential in treating COPD. Celastrol, an electrophilic triterpenoid extracted from the Thunder god vine (Trypterygium wilfordii), acts as an Nrf2 agonist and NADPH oxidase inhibitor. Despite Celastrol’s dual antioxidant mechanisms, NOX deficiencies may increase patients’ susceptibility to severe bacterial infection [68,80]. Hydrogen is a potent and safe antioxidant. Hydrogen-rich pure water has been found to decrease DNA oxidative stress in preclinical studies [68].

## 7. Clinical Trials

COPD is a complex disease that involves many molecular pathways in its pathogenesis. As described in this paper, most current pharmacological treatments are corticosteroids or bronchodilators that primarily alleviate symptoms associated with COPD or stabilize the disease. Current and future molecular approaches provide more options for targeted and personalized therapies for patients.

The current clinical trial scene for COPD treatment is still undergoing trials of evaluation and assessment. A little over 100 active trials were found on ClinicalTrials.gov using the “COPD Treatment” language search. Of these clinical trials, we identified eight drugs or cell therapy clinical trials currently being conducted for COPD treatment. Fexlamose is an interventional nebulizer solution with mucolytic properties studied in adults with moderate to severe COPD [81]. Tanimilast is a PDE4 inhibitor currently undergoing Phase III clinical trials (NCT04636801, NCT04636814), as is Tozorakimab, an IL33 inhibitor (NCT06040086). These molecular agents modulate specific targets essential in the pathogenesis of COPD, as discussed in the other parts of the paper (Table 6). 

The current state of COPD clinical trials also examines strategies that can repair tissue damage and reverse lung damage [82]. Regenerative therapies are appealing strategies. These therapies utilize exogenous stem cells to regenerate the functional deficit caused by COPD. Four other clinical trials utilize cell therapy. Preclinical studies found that human umbilical mesenchymal cells reduce lung emphysema and modulate inflammation [83]. Four phase I/II clinical trials are currently enrolling participants in studies evaluating the feasibility of mesenchymal stem cells (NCT04047810, NCT06491043, NCT05147688) and airway basal cells (NCT05638776).

Clinical trials require extensive time for completion, but exciting innovations are underway. We can look forward to new and approved treatment options for COPD patients.

## 8. Summary and Conclusions

COPD is a complex heterogeneous disease, and although targeting specific pathways may be conceptually attractive, it may not be sufficient to attenuate inflammation. The molecular pathogenesis of COPD has revealed that several pathways are activated, involving the recruitment of inflammatory cells such as neutrophils, lymphocytes, and eosinophils. A better understanding of the disease phenotype for individual patients with the contribution of specific cell types has provided the opportunity to personalize treatment approaches in COPD. While promising, biologics for COPD are still in the experimental stages, and more robust clinical trials are needed. Similarly, targeting specific inflammatory pathways will also need more clinical studies. Given the complex pathogenesis, a combinatorial approach to modulate multiple pathways may hold promise for the future.

## Figures and Tables

**Figure 1 ijms-26-02184-f001:**
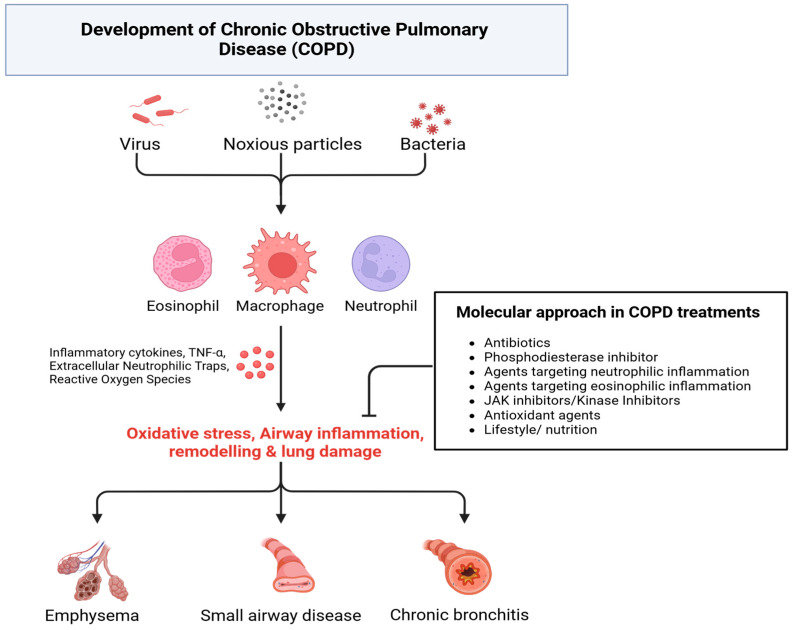
COPD Pathogenesis (created in https://BioRender.com).

**Table 1 ijms-26-02184-t001:** Summary of inhaled therapies for COPD.

Inhaled Therapies for COPD		
Specific	Mechanism	References
LABAs (Long-Acting Beta-Agonists)	Bronchodilation via beta-2 receptor activation. Improve dyspnea, reduce exacerbations, and improve health status Side effects: Tremors, tachycardia, nervousness, headache, dizziness	[17,18]
LAMAs (Long-Acting Muscarinic Antagonists)	Inhibit muscarinic receptors to cause bronchodilation. Improve lung function, reduce exacerbations Side effects: Dry mouth, cough, constipation, blurred vision	[17,18]
ICS (Inhaled Corticosteroid)	Anti-inflammatory effects. Reduces exacerbations, improves symptoms, and reduces mortality in some studies. Side effects: Oral thrush, hoarseness, pneumonia, systemic corticosteroid side effects	[1,19]

**Table 2 ijms-26-02184-t002:** Summary of phosphodiesterase inhibitors for COPD treatment.

Phosphodiesterase Inhibitors		
Specific	Mechanism	References
Theophylline	Nonspecific phosphodiesterase inhibitors that reduce inflammation, is a weak bronchodilator, decreased respiratory muscle fatigue, and has mild diuretic properties Side effects: Had limited utility due to side effects and drug interactions	[16]
Roflumilast/Tanimilast	Phosphodiesterase 4. Reduces airway inflammation via the cAMP pathway. Improves lung function and reduces exacerbations in severe COPD Side effects: weight loss, depression, gastrointestinal symptoms, insomnia	[29,30,31,32]
Ensifentrine	Phosphodiesterase 3 and 4. Dual bronchodilator and anti-inflammatory effects. Improves lung function and reduces exacerbations Side effects: Similar adverse events to placebo, low incidence of side effects	[39]

**Table 3 ijms-26-02184-t003:** Summary of Biologics Targeting Inflammation for COPD Treatment.

Molecules Targeting Inflammation		
Specific	Mechanism	References
Macrolides	Anti-inflammatory and immunomodulatory effects. Long-term usage reduces COPD exacerbations. Side effects: Associated with hearing loss, prolonged QT interval, bacterial resistance	[34,35,36]
Corticosteroids	Reduce inflammation by inhibiting inflammatory mediators. Alleviate inflammation and reduce exacerbations. Side effects: Increased risk of infections, osteoporosis, diabetes, weight gain	[44,45,46]
TNF-α inhibitor	Inhibit the action of TNF-α, reducing inflammation. Showed a reduction in COPD exacerbation and a potential synergy with corticosteroid therapy Side effects: Increased risk of infections, gastrointestinal issues	[47]
NET inhibitor	Modulate neutrophilic inflammation. Further in vitro/in vivo research is needed. Side effect: pharmacological inhibition of NETs attenuates inflammation while restoring lung function in mice model	[48]
Mepolizumab	Modulate eosinophilic inflammation. Target specific cytokines or receptors to reduce eosinophilic inflammation, IL-5. Reduce exacerbations in eosinophilic COPD; improve lung function and symptoms in specific phenotypes. Side effect: Further research/observation is needed	[43,49]
Dupilumab	Modulate eosinophilic inflammation. Target IL-4 and IL-13, which are the primary drivers of type 2 inflammation Side effect: Increased risk of infections, elevated liver enzymes, and potential cardiovascular effects	[43,49,50]
Lebrikizumab	Modulate eosinophilic inflammation. Target IL-13. Clinical trials reported no effects on lung function.	[43,49]
Benralizumab	Modulate eosinophilic inflammation. Target IL-5. Clinical trials reported minimal positive effects on lung function.	[43,49,51,52]
Itepekimab	Target IL-33. Phase 2 trials reported good improvement in lung functions; however, the study did not reach its endpoint aims.	[53]
Tozorakimab/Tezepelumab	Target IL-33. Clinical trials are undergoing	[54]

**Table 4 ijms-26-02184-t004:** Summary of Small-Molecule Inhibitors for COPD Treatment.

Small-Molecule Inhibitors			
	Specific	Mechanism	References
Other kinase inhibitors	Dilmapimod	P38 MAPK inhibitors. Currently in Phase 2, no significant efficacy	[63,64,65,66,67]
	Losmapimod		
JAK inhibitors	Tofacitinib	Target the JAK-STAT signaling pathway. Early phases of clinical trials, pending results. JAK inhibitors suppress the immune system, increasing the risk of infections, a significant concern for COPD patients already prone to respiratory infections	
	Ruxolitinib		
	LAS194046	In vitro study shows anti-inflammatory and antioxidant effects	

**Table 5 ijms-26-02184-t005:** Summary of Antioxidant Therapies for COPD Treatment.

Antioxidant Therapies			
Group	Specific	Mechanism	References
	N-Acetyl-L-cysteine	Increases cysteine’s stability and absorption for the synthesis of glutathione. Improve mucociliary clearance; reduce inflammation	[68,69,73]
A Thiol-based antioxidant	Procystein	A cysteine donor that improves the synthesis of cellular glutathione.	[74]
	Erdostein	Has mucoactive and antioxidant properties	[75]
	Carbocystein	A mucolytic antioxidant increases sialomucin content while significantly reducing the levels of pro-inflammatory Interleukin 6	[75]
	Fudostein	Has greater bioavailability that can increase cellular cysteine, reduce mucin production, and goblet cell hyperplasia.	[68]
Agonists of Nuclear factor erythroid 2-related factor 2	Imidazoline	Activate Nrf2 pathway to enhance antioxidant gene expression and reduce inflammation. Favorable results reported in pre-clinical studies	[68]
	Sulforaphane	Activate Nrf2 pathway to enhance antioxidant gene expression and reduce inflammation. Favorable results reported in pre-clinical studies	[68]
	Chalcones	Favorable results reported in pre-clinical studies	[68]
Glutathione Peroxidase	Ebselen	A mimic of glutathione peroxidase-1 catalyzes hydrogen peroxide and lipid hydroperoxide reduction. Protects against inflammation resulting from cigarette smoke	[68,76]
	Thioredoxin	A redox sensor that reduces oxidized proteins and enhances autophagy. Improve NF-kB activity by preventing it from translocating to the nuclease and subsequently activating inflammatory genes.	[69]
Agents from natural sources	Celastrol	Nrf2 agonist and NADPH oxidase inhibitor	[68]
	Hydrogen	Hydrogen-rich pure water has been found to decrease DNA oxidative stress in preclinical studies.	[68]

**Table 6 ijms-26-02184-t006:** Current clinical trials on COPD treatment.

Current COPD Clinical Trials		
Drug/Intervention	Mechanism	NCT Number
Fexlamose	mucolytic	NCT06731959
Mesenchymal stem cells	cell therapy	NCT04047810, NCT06491043, NCT05147688
Airway basal cells	cell therapy	NCT05638776
Tanimilast	PDE-4 inhibitor	NCT04636801, NCT04636814
Tozorakimab	IL-33 inhibitor	NCT06040086

## Data Availability

This review paper contains new data. Data sharing is not applicable to this article. The figure is generated on Biorender.com.

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
