# Peer review of "Molecular Approaches to Treating Chronic Obstructive Pulmonary Disease: Current Perspectives and Future Directions"

_ijms, 2025, doi:10.3390/ijms26052184_

Round 1

Reviewer 1 Report

Comments and Suggestions for Authors

this is an attempt to summarize the evidence on existing, investigational, and potential molecular approaches in COPD

overall the content is well organized and the choice of the pharmacological/biological classes well argumented

However I have some minor comments

You slightly overestimated  the "molecular" nature of some treatments in COPD so please exclude

1 antibiotics (macrolides) are able to modulate inflammation but their long-term use is COPD is still controversial due to related risk of hearing loss or of QT interval prolongation associated with long term use. You only mentioned azythromicine. However all macrolides exhibit such activities. So it would be appropriate to rather mention macrolides in fig 1 and in table 3. Also if you are aware of any macrolide which is a more potent antiinflammatory and a weaker antibiotic and which is also tested in copd, please refer it in this draft 

Also macrolides are not biologics so please correct

the rest is very good 

2 lifestyle and nutrition unfortunately do not have a documented and specific molecular target, there might be some prospect for if nutraceuticals are considered but it is not the case. so please delete or ammend the related information. molecular effect of a nutraceutical in copd would even be very interesting 

3 table 1 first line therapy should read first line inhaled therapy and should include the combinations onf LABA and LAMA and of LABA and inhaled corticosteroids, given you also cited triple combination in the table

Author Response

Response to Reviewer 1 Comments

1. Summary

Thank you very much for taking the time to review this manuscript. Please find the detailed responses below and the corresponding revisions/corrections highlighted in the re-submitted files.

2. Point-by-point response to Comments and Suggestions for Authors

Comments 1:

“1 antibiotics (macrolides) are able to modulate inflammation but their long-term use is COPD is still controversial due to related risk of hearing loss or of QT interval prolongation associated with long term use. You only mentioned azythromicine. However all macrolides exhibit such activities. So it would be appropriate to rather mention macrolides in fig 1 and in table 3. Also if you are aware of any macrolide which is a more potent antiinflammatory and a weaker antibiotic and which is also tested in copd, please refer it in this draft 

Also macrolides are not biologics so please correct the rest is very good.”

Response 1:

Thank you for the comment. We agree that all macrolides exhibit such activities and have

now mentioned macrolides rather than azithromycin. We are also not aware of a macrolide

that is considered more potent anti-inflammatory and weaker antibiotic. Therefore, we have

not added macrolides in this manuscript. Therefore, we have not added any more macrolides in this draft.

We modified Table 3 to mention “Macrolides” instead of “Azithromycin”

We renamed the Biologics targeting inflammation Table 3 to Molecules targeting inflammation

Comments 2:

2 lifestyle and nutrition unfortunately do not have a documented and specific molecular target, there might be some prospect for if nutraceuticals are considered but it is not the case. so please delete or ammend the related information. molecular effect of a nutraceutical in copd would even be very interesting 

Response 2: We agree that lifestyle and nutrition do not have a specific molecular target.

Thus, we have accordingly, deleted nutrients as a target.

Comments 3:

3 table 1 first line therapy should read first line inhaled therapy and should include the combinations onf LABA and LAMA and of LABA and inhaled corticosteroids, given you also cited triple combination in the table

Response 3: Thank you for your suggestion. We simplified things by taking out the combination triple therapy row and instead making that row "ICS."

Reviewer 2 Report

Comments and Suggestions for Authors

The authors summarize the literature, clinical trials, and experimental therapeutic approaches regarding COPD. The review is comprehensive and well-written, and besides providing a brief update on cotemporally (LABA, LAMA, corticosteroid) treatments, it discusses promising therapies in the phase of clinical trials and preclinical (experimental) phase. This review article uses 79 references, which are all relevant to COPD present and possible future therapy.

Critics:

  • Please list genetic abnormalities (i.e., Alph-1 Antitrypsin Deficiency) as a cause of COPD. Although genetic abnormalities are less frequent compared to acquired ones, these pathologies are part of the highly diverse roots of COPD.
  • In line 274, Please correct the name to Nicotinamide Adenine Dinucleotide phosphate (NADPH)oxidases. Hydrogen is not part of the enzyme name in any meaning. NOX is an enzyme family that consists of 7 enzymes discovered to date. Please discuss it as an enzyme family and not as a single enzyme.
  • In lines 276-277, the authors state: “In COPD patients, NOX’s levels and activity are lower compared to healthy patients, which results in the oxidant-antioxidant imbalance and increases their susceptibility to bacterial infection 68,69.” The here-referenced manuscripts do not state that COPD patients demonstrate lower NOXs levels or activity. Reference 68 only discusses mouse models deficient for NOX, which are protective against oxidative stress. However, patients with NOX deficiency carry an increased susceptibility to life-threatening bacterial infections. Please correct this statement accordingly.
  • Along the paragraph discussing antioxidant status and therapy, glutathione is capitalized in multiple cases (“Glutathione”). Please use lowercase as glutathione.
  • In lines 292-293, the authors state: “Erdostein and Procysteine, when used together, have been shown to improve ROS production and inflammation cytokines in smoker patients with COPD.” I believe the authors meant to say that erdosterin and procysteine decrease ROS levels, not improve ROS production. Please correct this sentence accordingly.
  • In line 304, Please correct the typo if the author refers to the gene's name here. The correct spelling is NRF2.
  • In lines 312-313, the authors state: “Another key antioxidant defense strategy is facilitated by the enzyme Glutathione Peroxidase (GPx).” Glutathione Peroxidases belong to an enzyme family that consists of 8 enzymes. Please use plural forms in the enzyme name or define individual isoforms.
  • In lines 313-314, the authors state: “GPx catalyzes hydrogen peroxide and lipid hydroperoxide reduction 68,75; however, GPx is reduced in COPD patients 75.” According to the referenced manuscript, GPx-1 levels were significantly lower, while GPx-3 did not change. Please correct the sentence accordingly.
  • Please add reference 68 to Table 5 on the Celastrol row.

Author Response

Response to Reviewer 2 Comments

1. Summary

Thank you very much for taking the time to review this manuscript. Please find the detailed responses below and the corresponding revisions/corrections highlighted in the re-submitted files.

2. Point-by-point response to Comments and Suggestions for Authors

Comments 1:

  • Please list genetic abnormalities (i.e., Alph-1 Antitrypsin Deficiency) as a cause of COPD. Although genetic abnormalities are less frequent compared to acquired ones, these pathologies are part of the highly diverse roots of COPD.

Response 1: Thank you for pointing this out. We agree with this comment. Therefore, we have added more discussion on the matter

The updated text can be found in red on page 2, line 42

“Genetic risk factors, most notably mutations in the SERPINA 1 gene causing alpha1-antitrypsin deficiency 8, are associated with an increased susceptibility to COPD; other genetic risk loci have also been identified, such as CHRNA3, FAM13A, HHIP, and RIN3 9

Comments 2:

  • In line 274, Please correct the name to Nicotinamide Adenine Dinucleotide phosphate (NADPH)oxidases. Hydrogen is not part of the enzyme name in any meaning. NOX is an enzyme family that consists of 7 enzymes discovered to date. Please discuss it as an enzyme family and not as a single enzyme.

Response 2: Thank you for pointing this out. We have, accordingly, revised NADPH naming and referring to NOX as a family of enzymes to emphasize this point. This change can be found on page 8 line 277

Comments 3:

  • In lines 276-277, the authors state: “In COPD patients, NOX’s levels and activity are lower compared to healthy patients, which results in the oxidant-antioxidant imbalance and increases their susceptibility to bacterial infection 68,69.” The here-referenced manuscripts do not state that COPD patients demonstrate lower NOXs levels or activity. Reference 68 only discusses mouse models deficient for NOX, which are protective against oxidative stress. However, patients with NOX deficiency carry an increased susceptibility to life-threatening bacterial infections. Please correct this statement accordingly.

Response 3: We agree with your comment. We have, accordingly, revised to emphasize this point. Please see below the details:

We discussed NOX’s role in generating endogenous ROS on page 8 line 279

The superoxide anions produced by NOX can be converted rapidly into hydroxyl radicals and/or H2O271.”

We discussed the levels/activity of NOX in advanced COPD disease progression and smoked-induced COPD. All in all, the level/activity of NOX is high in COPD disease on page 8 line 279.

Added “It is reported that NOX1, NOX2, NOX4 and NOX5 are active in patients with end-stage COPD, while NOX1 and NOX4 are primarily detected in smoke-induced COPD72

Delete “In COPD patients, NOX’s levels and activity are lower compared to healthy patients, which results in the oxidant-antioxidant imbalance and increases their susceptibility to bacterial infection 68,69

We iterate NOX deficiency may increase patients susceptibility to bacterial infection on page 9.

Comments 4:

  • Along the paragraph discussing antioxidant status and therapy, glutathione is capitalized in multiple cases (“Glutathione”). Please use lowercase as glutathione.

Response 4: Thank you for pointing it out. We have, accordingly, revised “Glutathione” to “glutathione”

Comments 5:

  • In lines 292-293, the authors state: “Erdostein and Procysteine, when used together, have been shown to improve ROS production and inflammation cytokines in smoker patients with COPD.” I believe the authors meant to say that erdosterin and procysteine decrease ROS levels, not improve ROS production. Please correct this sentence accordingly.

Response 5: Agree. That sentence initially means to say that erdostein and procystein, together, improved smoked-induced COPD outcomes by way of better ROS reduction. We have, accordingly, revised that sentence to emphasize this point.

The change can be found on page 8 line 299

The new sentence is “Erdostein and Procysteine, when used together, have been shown to reduce ROS production and inflammation cytokines in smoker patients with COPD 73.

Comments 6:

  • In line 304, Please correct the typo if the author refers to the gene's name here. The correct spelling is NRF2.

Response 6: The transcription factor’s name is modified to Nrf2, page 8, line 311

Comments 7:

  • In lines 312-313, the authors state: “Another key antioxidant defense strategy is facilitated by the enzyme Glutathione Peroxidase (GPx).” Glutathione Peroxidases belong to an enzyme family that consists of 8 enzymes. Please use plural forms in the enzyme name or define individual isoforms.

Response 7:

Thank you for pointing this out. We have made changes accordingly. A sentence was added to mention GPXs is a family of 8 enzymes. Wherever applicable, we mention the specific homolog. If not, we use the plural form for GPXs.

The changes can be found on page 9 line 319

The new sentence is: Another key antioxidant defense strategy is facilitated by Glutathione Peroxidases (GPXs), which is a group of 8 enzymes that catalyze hydrogen peroxide and lipid hydroperoxide reduction 68,75

Comments 8:

  • “In lines 313-314, the authors state: “GPx catalyzes hydrogen peroxide and lipid hydroperoxide reduction 68,75; however, GPx is reduced in COPD patients 75.” According to the referenced manuscript, GPx-1 levels were significantly lower, while GPx-3 did not change. Please correct the sentence accordingly.”

Response 8:  We agree with your comment. We have, accordingly, revised the sentence to emphasize the point that reference 75 reported GPx-1, specifically, is reduced in COPD patients compared to non-COPD patients.

We also include one more reference, which studies the activity of the GPX family moderate versus severe COPD patients.

The changes can be found in on page 9 line 319

The new sentence is: It is found that GPXs activity was significantly lower in patients with severe COPD compared to patients with moderate COPD 76. A recent study has determined the specific homolog GPx-1 is significantly reduced in COPD patients compared to non-COPD patients 75

Comments 9: “Please add reference 68 to Table 5 on the Celastrol row”

Response 9: We have, accordingly, added reference to the Celastrol row in Table 5